# Evaluation of Behavioral Changes and Tissue Damages in Common Carp (*Cyprinus carpio*) after Exposure to the Herbicide Glyphosate

**DOI:** 10.3390/vetsci8100218

**Published:** 2021-10-05

**Authors:** Ahmad Mohamadi Yalsuyi, Mohammad Forouhar Vajargah, Abdolmajid Hajimoradloo, Mohsen Mohammadi Galangash, Marko D. Prokić, Caterina Faggio

**Affiliations:** 1Department of Aquaculture, Faculty of Fisheries and Environment, Gorgan University of Agricultural Sciences and Natural Resources, Gorgan 4913815739, Iran; ahmad_m.yalsuyi@yahoo.com (A.M.Y.); A_hajimoradloo@gau.ac.ir (A.H.); 2Department of Fisheries, Faculty of Natural Resources, University of Guilan, Sowmehsara 4199613776, Iran; Mohammad.forouhar@yahoo.com; 3Department of Environmental Sciences and Engineering, Faculty of Natural Resources, University of Guilan, Sowmehsara 4199613776, Iran; m_mohammadi@guilan.ac.ir; 4Department of Physiology, Institute for Biological Research “Siniša Stanković”, National Institute of Republic of Serbia, University of Belgrade, 11060 Belgrade, Serbia; marko.prokic@ibiss.bg.ac.rs; 5Department of Chemical, Biological, Pharmaceutical and Environmental Sciences University of Messina, 98122 Messina, Italy

**Keywords:** acute stress, glyphosate, behavioral responses, tissue damages, *Cyprinus carpio*

## Abstract

Pesticides can induce changes in behavior and reduce the survival chance of aquatic organisms. In this study, the toxic effects of glyphosate suspension (Glyphosate Aria 41% SL, Tehran Iran) on behavior and tissues of common carp (*Cyprinus carpio*) were assessed. For this purpose, a 96 h LC_50_ of glyphosate suspension (68.788 mL·L^−1^) was used in the toxicity test. All individuals were divided into control and treatment groups with four replicates. Exposure operations were performed under two conditions: increasing concentration of suspension from 0 to 68.788 mL·L^−1^; then, decreasing to the first level. The swimming pattern was recorded by digital cameras during the test and tissue samples were collected at the end of the test. There were significant differences between the swimming pattern of treated individuals and control ones during both steps. The sublethal concentration of glyphosate led to hypertrophy, hyperplasia and hyperemia in the gill of fish. However, changes were obvious only after sampling. The exposed fish also displayed clinical signs such as darkening of the skin and increasing movement of the operculum. Moreover, glyphosate suspension affected swimming patterns of fish suggest that the swimming behavior test can indicate the potential toxicity of environmental pollutants and be used as a noninvasive, useful method for managing environmental changes and assessing fish health conditions by video monitoring.

## 1. Introduction

The projection is that the world population will excess seven billion people and will reach nine billion by 2050 [1]. Population growth will overcome agriculture product growth by 2030. A series of factors (land degradation, falling cropland per person, global climate change, water crisis, uneven distribution of resources and reduced yields per hectare) in some continents and areas (i.e., Africa and west Asia) of the world already threaten human food safety [2,3,4]. Plant pests are one of the important contributing factors in reducing agriculture production. Global statistics show that an average of 35% of agricultural products is lost due to the effects of preharvest pests [5,6]. However, the use of pesticides appears to become less effective on crop pests [5]. As an example, Raven et al. [7] showed that even though agriculture production and use of pesticides significantly increased (about 33%) after the 1940s in the USA, the percent of agricultural waste lost because of pests has not changed significantly. The fold increase went up to 15–20 in recent years [5,8] and the total annual cost spent by farmers to supply and buy pesticides is about USD 40 billion [9]. It was observed that pesticides near the target organism, affect a large number of the nontarget organisms. These effects are not limited only to the death of the nontarget organisms, but are also linked to changes in organism fitness (reproduction, survival and longevity) [10,11,12,13,14,15,16]. Environmental pollutants can affect behavior and life history parameters including growth and reproductive functions of nontarget organisms. Thus, there are serious concerns about the increased use of pesticides as a way to increase crop yield [17,18,19].

Among herbicides, glyphosate (C_3_H_8_NO_5_P—41% SL) is one of the most commonly used. In recent years, due to its impressive effect on weeds, the use of glyphosate has rapidly increased worldwide, especially in developing countries [20,21]. Glyphosate is sold in over 100 countries and its global use reached more than 800,000 tons in 2014 [22]. It is a type of herbicide that displays an effect on the activity of 5-enolpyruvylshikimate-3-phosphate synthase (EPSP) and inhibits or impairs the synthesis of aromatic amino acids in plant cellulose [20,23]. Aquatic environments are often the last destination for this pollutant [24,25,26], and due to its high solubility and relatively long half-life, it is usually present in water ecosystems. It leaches into water through several routes such as rainfall, river and soil erosion [27]. The growing trend in its usage has led to an increase in residues and adverse effects of this herbicide on biotic and abiotic components of the ecosystems [28]. The results of previous studies pointed out that glyphosate can be toxic to nontarget organisms. Evans et al. [16] studied the toxicity effect of glyphosate-based herbicides on the behavior and survival rate of nontarget organisms (spiders and ground beetles). Results of their study showed that glyphosate suspension, apart from affecting arthropod community dynamics, can significantly change behavior (prey behavior) and reduced the survival chance of spiders and ground beetles. In this study we used the common carp (*Cyprinus carpio*) as a model organism. Exposure of the common carp to glyphosate can reduce the rate and survival chance [29,30,31,32].

There are various methods to evaluate the possible toxicity of the pesticide. Most of them require sacrificing a great number of individuals, especially for the LC_50_ 96 h test, which is based on finding the lethal concentration that kills half of the population [18]. Histopathological testing on the other hand has some limitations, such as: time-consuming processes for sampling, fixation and preparing of tissue sections; need to use different chemical substance and relatively expensive tools [33]. Behavioral tests are a noninvasive method for evaluation of adverse effects of stressor parameters and do not require killing fish or the use of expensive tools [34,35]. Behavior is the result of the interaction between internal and external stimuli [36]. Hence, behavioral studies are based on behavioral changes of the organism that are exposed to a wide range of stressor parameters such as pollutants, prey, hunter, acute or chronic changes of water physicochemical parameters and pathogens [37,38]. Finally, this type of studies can provide useful and new information about the effects of pollutants on an organism that could not be found with other methods [36,39].

The present study aims to evaluate behavioral changes and tissue damages of common carp (*Cyprinus carpio)* exposed to sublethal and lethal concentrations of glyphosate. We also investigate the possible use of the swimming pattern of fish as a behavior parameter in toxicological studies and environmental quality assessment [34].

## 2. Materials and Methods

All steps of the present study were performed according to ethical standards and valid regulations. The protocol of the study was according to guidelines issued by the Gorgan University of Agricultural Sciences and Natural Resources Research Ethics Committee (No. IR-GAUEC207s-2020).

### 2.1. Preparing

For the experimental procedure, 100 fingering common carp (*Cyprinus carpio*) with an average weight of 4.85 ± 0.6 g were bought and transferred to the research center (laboratory of the Faculty of Natural Resources, University of Guilan, Guilan province, Iran) The fish were randomly divided into 4 tanks (250 L— 25 fish in each tank). In order to adapt to the laboratory conditions, the fish were maintained in these tanks for 2 weeks. They were fed a commercial diet (produced by Faradaneh Co., Tehran, Iran) at 3% of biomass weight, twice a day, during the adaptation period. Physicochemical parameters of water were measured every day and they were the same in all tanks (pH 6.7–7.4, temperature 25 ± 1 °C, DO 8 mg·L^−1^, NH_3_ < 0.02 mg·L^−1^ and total hardness 185 mg CaCO_3_). The pH and temperature, dissolved oxygen (DO), NH_3_ concentration and total hardness of water (CaCO_3_ concentration) were measured by a digital soil and substrate pH meter (S500 pro, Aqua Masters, Burbank, CA, USA), a dissolved oxygen meter for aquaculture (HI9147, HANNA Instruments, Bertoki, Slovenia) and multiparameter photometers (7100, Palintest Co., Gateshead, UK) twice a day, respectively.

### 2.2. Toxicity Test

According to Vajargah et al. [8] and Hedayati et al. [18], after the adaptation period, 60 fingerling fish (average of weight 4.85 ± 0.62 g) were divided into 4 groups with 3 replicates (12 aquariums 100 × 40 × 50 cm) and exposed to 4 concentrations of commercial formulation of glyphosate suspension (glyphosate Aria 41% SL, Tehran, Iran). Nominal concentrations of glyphosate were 0, 50, 100 and 150 mL·L^−1^ and the test duration was 96 h. The fish mortality rate was calculated at 24, 48, 72 and 96 h after exposure. Fish were transferred into the test tank 16 h before beginning the test and they did not feed during the toxicity test. Water physicochemical parameters were kept the same as one of the adaptation periods (pH 6.7–7.4, temperature 25 ± 1 °C, DO 8 mg·L^−1^, NH_3_ <0.02 mg·L^−1^ and total hardness 185 mg CaCO_3_). The LC_50_ 96 h test was a static system. Finally, glyphosate concentrations were added manually and the pesticide was distributed by water circulation inside the aquarium.

### 2.3. Histopathological Test

The gill samples of fish were collected at 24, 48, 72 and 96 h after exposing the fish to glyphosate (one sample for each replication) and they were fixed by diluted Formalin solution (Formaldehyde 10% *v*/*v*, Sigma^®^, St. Louis, MO, USA). Formalin of the gill samples was replaced 24 h after sampling [8]. The second gill arch from the fish’s left side was selected for sampling. The samples were placed in a series of alcohols (50, 70, 80 and 96%) for half an hour. Immediately after that, the gill sample was washed with 1-butanol alcohol (for 2 h) and then they were placed into chloroform for clarifying for 1 h. After this step, the gill samples were placed into an incubator at 37 °C for paraffinization and softening using a solution of chloroform and paraffin (1:1). Then, samples were incubated in pure paraffin at 54 °C and were prepared for tissue incisions after cooling. The tissue incisions were obtained with an automatic tissue processor machine (TP1020, Leica Microsystems Inc., Buffalo Grove, IL, USA) and their thickness was 6 µm. The tissues incisions were stained by hematoxylin─eosin [33]. Tissue damages were observed and evaluated by light microscopy (Model RH-85 UXL, UNILAB^®^, Mumbai, India). Organ damages were analyzed according to Vajagah et al. [8] and compared with each other.

### 2.4. Behavioral Test

According to Kang et al. [34] and Kane et al. [36], after measurement of the 96 h LC_50_ of the glyphosate suspension (glyphosate Aria 41% SL, Tehran, Iran) the behavior test was performed. The selection of fish was based on a completely random design, namely: 8 fingering common carp (*Cyprinus carpio*) with an average weight of 5.93 ± 0.8 g were selected and divided into 2 groups (control and glyphosate treatment) with 4 repetitions and maintained in 8 test tanks (the volume of water was 5 L in the test tank, one fish in each tank). Fishes were transferred into the test tanks 12 h before beginning the behavior test for adaptation and were not feed during the behavioral test. The test tanks had a water inlet and an outlet, and the water flow rate was the same in these tanks. The rate of water flow was chosen so that the nominal concentration of the commercial formulation of glyphosate could reach 96 h LC_50_ at 12 h. The water flow rate was 416.667 mL·h^−1^. For the glyphosate treatment, the nominal concentration of suspension was increased from 0 mL·L^−1^ to the 96 h LC_50_ during the first step (12 h). Finally, in order to evaluate behavioral responses of fish to a return or improvement of environmental parameters, the nominal concentration of suspension was reduced from the 96 h LC_50_ to 0 mL·l^−1^ (12 h) during the second step (Figure 1b). The water flow rate of the control group tank was similar to the glyphosate treatment. In addition, water physicochemical parameters were similar to the adaptation period (pH 6.7–7.4, temperature 25 ± 1 °C, DO 8 mg·L^−1^, NH_3_ <0.02 mg·L^−1^ and total hardness 185 mg CaCO_3_). The nominal concentration of the commercial formulation of glyphosate was 0 mL·L^−1^ during both steps of the behavioral test. The water flow and the glyphosate concentration were maintained constant during the test by adjustable valves and a mixer and a precision pump (BT300-2J medium flow rate peristaltic pump, Longer Precision Pump Co., Ltd., Baoding, China), respectively. The water surface was fixed in the test tanks and water physicochemical parameters were checked every 6 h. Finally, all of the analyses were conducted in one day.

The swimming patterns of fish were recorded by 8 digital cameras (Canon, SX230 Hs, 5.0–70 mm) at 26 time points, 1 min apart (Figure 1c). Each camera was on top of a test tank (Figure 1a). The height of the cameras was 10 cm from the water surface. The total time of the recording parts for each fish was 26 min (26 separate 1 min pieces) and their format was MP4 (.mp4)

### 2.5. Data Analysis

The lethal concentration of glyphosate at intervals of 24, 48, 72 and 96 h (24 h, 48 h, 72 h and 96 h LC_50_ of glyphosate) were estimated through probit tests with a 95% confidence. To find the correlation between different nominal concentrations of commercial formulations of glyphosate and mortality we used a Spearman test (2-tail).

The video data were analyzed by Adobe After Effects software (AAE CS6) on a Windows platform (Windows 7 Ultimate, Microsoft corporation©, Redmond, WA, USA). This software program converted selected video data (resolution 640 × 480, 30 frames per second) to 1 frame per second and indicated the position of the fish in each frame through the location of the fish head (Figure 1a). The output file format of this software was a FLV video file (.flv). The total movement and distance from the center in different sections were measured by Digimizer (Version 4.6.1, MedCalc Software, Ostend, Belgium) on a Windows platform [40]. The differences between average measured indicators of treatment at different concentrations and the control group were calculated through a LSD test with a 95% confidence by SPSS software (SPSS Statistics 20, IBM, Armonk, NY, USA). Finally, the correlations between swimming parameters were checked with Pearson’s and Spearman’s two-tailed significance tests using SPSS software (SPSS Statistics 20, IBM).

The clinical signs of fish were reported by direct observation of recorded videos, counting of the average movements of the gill operculum in 1 min and a comparison of the color of the object (fish) for a period of time by Adobe After Effects software (AAE CS6, Adobe Co., Mountain View, CA, USA).

Behavioral changes of fish were characterized through the analysis of the average swimming speed (A.S.), total movement (T.M.), percent movement (P.M.), fastest movement (F.M.), average angular change of movement (A.C.) and the average distance from the center (D.C.); these parameters of swimming patterns were selected and modified according to Kane et al. [37] and Yalsuyi et al. [40] (Table 1).

## 3. Results

### 3.1. Results of the 96 h LC_50_ (Lethal Concentration of 50% of the Population in 96 h) Test

No mortality of fish was observed during the adaptation period. The results of the toxicity test showed that there was a significant correlation between the mortality rate of fish and the concentration of the commercial formulation of glyphosate (*p* < 0.01). Mortality was observed in all treatment steps (except for the control group at 0 mL·l^−1^ of glyphosate) and there were significant differences between mortality rates and treatment levels (*p* < 0.01). The 96 h LC_50_ of commercial formulation of glyphosate was 68.788 mL·L^−1^ and its 24, 48 and 72 h LC_50_ were 202.132, 130.014 and 92.798 mL·L^−1^, respectively. According to the results of the test of homogeneity of variance (Levene’s Test), we did not observe significant differences between the variance of groups (*p* > 0.05).

#### Results of the Histopathological Assay

We did not find any significant tissue damages in the control group. Damages were observed in all treatments of glyphosate (50, 100 and 150 mL·L^−1^) at 96 h after exposure. There were significant correlations between tissue damages and glyphosate concentrations (*p* < 0.05). The tissue damages including hyperemia, hypertrophy, hyperplasia, secondary lamellar adhesion, hemorrhage and necrosis were found in gill samples of fish that were exposed to sublethal and lethal concentrations of glyphosate (Table 2). The highest damages of gills were reported at 150 mL·L^−1^ of glyphosate. The gills hyperplasia (HP), hypertrophy (HT), swollen primary gill (SPG), secondary lamellar adhesion (SLA), hyperemia (H), hemorrhage (HR) and necrosis (N) were clearly observed in the samples (Figure 2).

The primary tissue damages were seen 24 h after exposure and they intensified during the 96 h LC_50_ test. Finally, we did not see or record any significant tissue damages in the control group (concentration 0 mL·L^−1^ of glyphosate suspension). The results of the histopathological assay of the gills are shown in Table 2.

### 3.2. Results of the Behavioral Assay

#### 3.2.1. Control Group

We did not record any mortality in the glyphosate treatment and control groups during the behavioral test. There was a significant correlation between average swimming speeds and total movements (*p* < 0.01). However, there were no significant correlations between these parameters and other swimming parameters (*p* > 0.05). In addition, there were no significant differences between swimming parameters at different time points and the test steps (Figure 3). Finally, no significant differences between the swimming patterns of the control group fish during the test steps were observed (Table 3).

The results showed that fish rarely changed their swimming directions and they were swimming near the middle of the test tanks in a circular path during the test (Figure 4a). The area of the test tank was 660.185 cm^2^ and the total movement of the fish in 1 min was usually less than half of the area of the test tanks.

#### 3.2.2. Glyphosate Treatment

During the behavioral test, no mortality was reported. There were significant correlations between swimming patterns parameters, the strongest correlation was between average swimming speed and total movement (*p* < 0.01); also, these two parameters had correlations with the fastest movement (*p* < 0.01), average angular changes of movement (*p* < 0.01), the average distance from the center (*p* < 0.01) and percent movement (*p* < 0.05). There were significant differences between the swimming pattern parameter of fish in the same concentration of different steps (*p* < 0.05). However, these differences were not seen at all concentrations (Figure 5).

Significant differences between swimming parameters of the glyphosate treatment and the control group were seen (*p* < 0.05). There were no significant differences between swimming parameters of the treatment group in step 1 and step 2 (Table 3). However, the fish were usually swimming near the water outlet of the test tank during step 1, while they tended to be near the water inlet during step 2 (Figure 4).

The slopes of the diagrams of swimming parameters of fish (treatment group) were positive in step 1, but their values and slopes decreased and became negative in step 2, respectively. However, these reductions were not significant at the beginning of step 2, which coincides with the reduced concentrations of glyphosate (Figure 6). The average values of parameters of the treatment group were usually higher than those of the control group during the test, except for the average value of the percent of movement, which was lower than that of the control group at the end of step 2 (Figure 6d).

The total movement of fish after exposure to glyphosate was significantly increased, with a value lower than half of the test tank area (660.185 cm^2^). Fish that were exposed to sublethal and lethal concentrations of commercial formulation of glyphosate showed clinical signs, such as darkening of the skin, increasing movement of the operculum, anxiety, jumping out of the water and swimming near the surface.

## 4. Discussion

Contaminants can induce adverse physiological, behavioral and histopathological effects on aquatic organisms over time, depending on various factors, such as concentration and chemical structures [41,42,43,44,45,46,47,48,49]. Results of the present study showed that sublethal and lethal concentrations of commercial formulation of glyphosate significantly reduce the survival rate of fish, and can lead to significant tissue damages, such as hyperplasia of fish gills. The toxic effects of glyphosate were also seen and recognized through changes in the fish swimming patterns. Behavioral changes can further influence survival chances, reproductive success, nutrition and growth behaviors of the organism [35,36,37,38].

Vajargah et al. [19] stated that the 96 h LC_50_ of glyphosate for fingerling common carp (*Cyprinus carpio*) was 92.71 mL·L^−1^ and this concentration was inconsistent with the result of the present study (68.788 mL·L^−1^). Nonetheless, all parameters in the present study were very similar to those in their study; differences were very small (nonsignificant) and were seen in the fish sizes. Body size is one of the intraspecific characteristics that could be related to differences in lethal concentration [37]. The average weight of fish in the present study and their study differed, and were 4.85 ± 0.62 and 7 ± 0.8 g, respectively. Thus, results of the 96 h LC_50_ test are limited as they reflect laboratory conditions, but they can be useful to determine a range of lethal concentrations of pollutants.

Glyphosate (C_3_H_8_NO_5_P 41% SL) acts on the activity of 5-enolpyruvylshikimate-3-phosphate synthase (EPSP) and inhibits or impairs the synthesis of the aromatic amino acids in plant cells [21,23]. The results of the present study showed that glyphosate displayed significant toxic effects on common carp (*Cyprinus carpio*) as an aquatic nontarget organism, inducing histopathological damages [50]. The results of Neskovic et al. [51] showed that exposure to 5 mg·L^−1^ of this pesticide can lead to epithelial hyperplasia, hypertrophy of chloride cells and lifting and rupture of the respiratory epithelium of carp gills. The results of their study were similar to ours. Tissue damages including hyperemia, hypertrophy, hyperplasia, secondary lamellar adhesion, hemorrhage and necrosis were found in gill samples of fish that were exposed to sublethal and lethal concentrations of glyphosate. There was a dose-dependent relationship between glyphosate concentrations and observed damages. Webster and Santos [52] stated that Roundup (common formulation of glyphosate) led to different variations in the complex interacting signaling pathways of juvenile female brown trout that control cellular stress response, particularly in apoptosis. In addition, the result of their study showed increased cell proliferation, cellular turnover and an up-regulation of metabolic processes.

After glyphosate exposure, the operculum movement (ventilatory frequency) and swimming activity of the fish was increased in a study conducted by Sinhorin [53]. Those results were in concordance to ours. In the present study, the swimming parameters, including average swimming speed, total movement, percent movement, fastest movement, average angular changes of movement and the average distance from the center, were changed and their values were significantly higher in the treatment group than in the control group. Moreover, the ventilatory frequency of fish was increased after glyphosate suspension exposure. Glyphosate has a toxic effect on the activity of the acetylcholine esterase enzyme and oxidative stress in common carp [54]. Results from the same authors showed the repression effect of glyphosate on acetylcholine esterase activity in the brain and muscles of fish and that its oxidative stress can lead to anxiety, increased fish metabolism, fatigue of fish and decreased energy levels. We showed that individuals exposed to glyphosate had clinical signs such as anxiety, and their percent movement significantly decreased at the end of step 2.

Behavior responses (such as swimming performance) as a tertiary level of physiological responses to a stressor can be used as a biomarker of stress [55]. Monitoring of the stressor factors in the environment is important [36], because these factors reduce organisms’ fitness and survival chance [56,57,58,59]. The change of fish swimming patterns was observed in the present study. When the glyphosate suspension concentration was increased, the swimming speed increased, with frequent changes in the direction of swimming, and the fish tended to be near the water outlet; conversely, a decrease in the concentration of stressor caused the fish to swim near the water inlet.

## 5. Conclusions

Results of the given study clearly showed that glyphosate affects the swimming patterns of fish and can induce histopathological tissue damages. Some behavioral parameters such as swimming speed, average angular change of movement, the average distance of fish from the tank’s center and the tendency of fish to be near the water inlet or outlet clearly point to anxiety in the fish and its tendency to escape contamination. Our results suggest that fish swimming parameters may be useful indicators of aquatic environments, and unlike some monitoring methods of pollutants effect (such as the 96 h LC_50_ test) they do not require killing fish, histopathological processes and fish catching. However, the data of the present study were limited to laboratory conditions and further studies are required. Swimming patterns were shown to be a promising method for collecting clinical signs of fish as a response to environmental stressors (such as pollutants).

## Figures and Tables

**Figure 1 vetsci-08-00218-f001:**
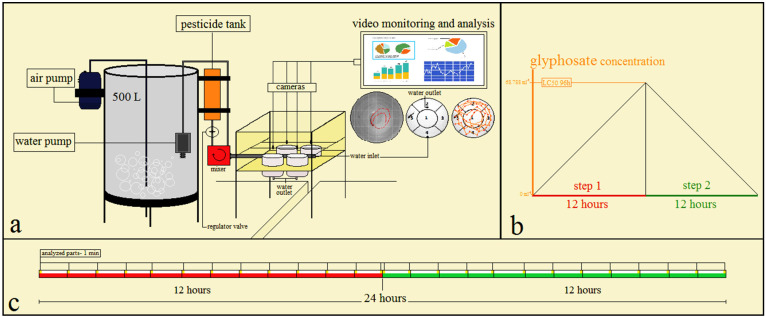
Schematic description of the test environment and its stages. (**a**) System design and components of the test. (**b**) Steps of the test; concentration of glyphosate suspension was increased and reached the 96 h LC_50_ during step 1, then was decreased and reached 0 mL·L^−1^ during step 2. (**c**) The timeline of the behavioral test: each step was 12 h and there were 26 time points. The behavior test was a dynamic test. The fish swimming was recorded at these time points of 1 min duration. The water flow during the test was 416.667 mL·h^−1^.

**Figure 2 vetsci-08-00218-f002:**
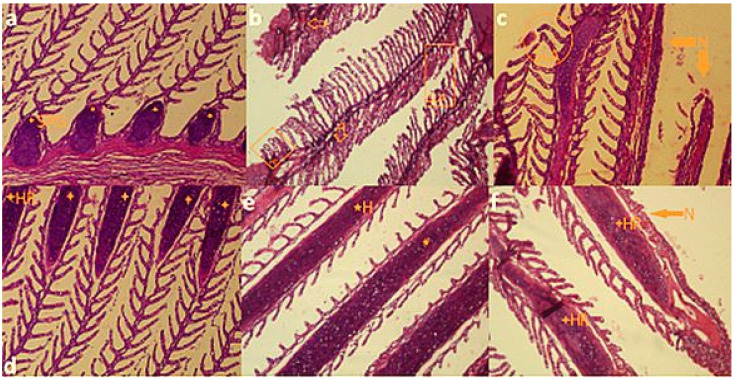
Photomicrographs of common carp (*C. carpio*) after 96 h exposure to lethal concentrations of commercial formulation of glyphosate (glyphosate Aria 41% SL suspension): (**a**) swollen primary gill (SPG) in 50 mL·L^−1^; (**b**) hyperplasia (HP), hypertrophy (HT), secondary lamellar adhesion (SLA) and necrosis (N) of gills in 100 mL·l^−1^; (**c**) secondary lamellar adhesion (SLA) and necrosis (N) of gills in 150 mL·L^−1^; (**d**,**e**) hemorrhage (HR) and hyperemia (H) in 100 and 150 mL·L^−1^, respectively; (**f**) hemorrhage (HR) and necrosis (N) of gills in 150 mL·L^−1^. All pictures are magnified ×10.

**Figure 3 vetsci-08-00218-f003:**
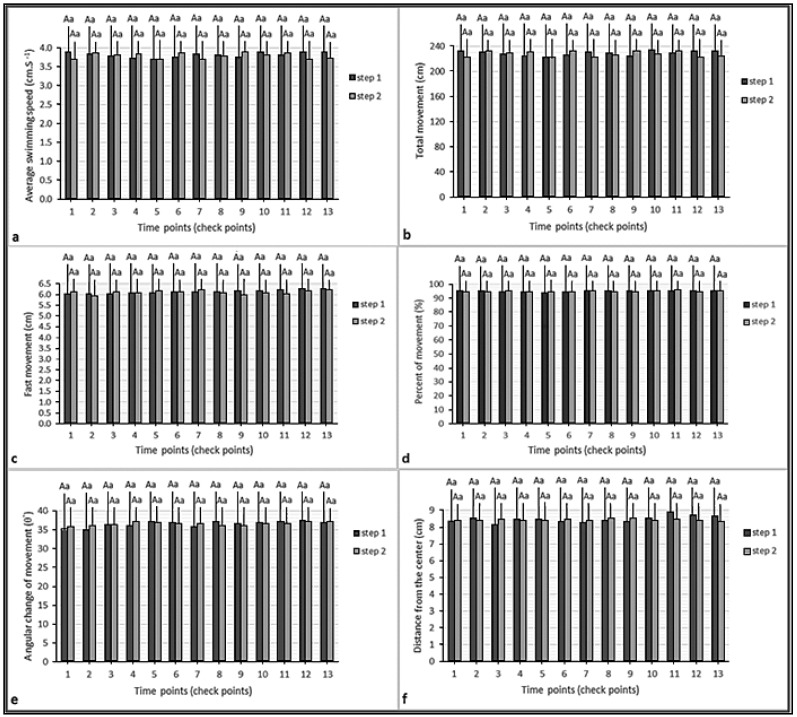
Parameters of swimming pattern of the fish in the control group: (**a**) average speed; (**b**) total movement; (**c**) fast movement; (**d**) percent of movement; (**e**) average angular change of movement; (**f**) distance from the center. Similar lowercase letters (a) indicate nonsignificant differences between values of the same color columns (*p* > 0.05). Similar uppercase letters (A) indicate nonsignificant differences between values of paired columns (*p* > 0.05).

**Figure 4 vetsci-08-00218-f004:**
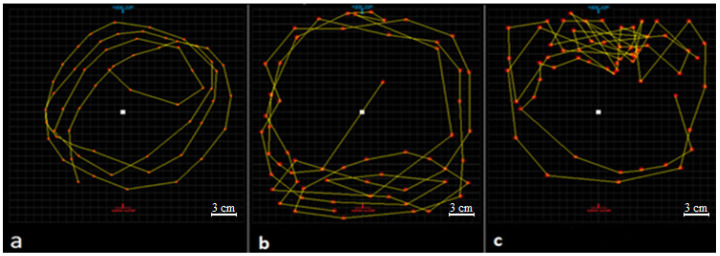
Fish swimming paths at different steps of the test (1 min): (**a**) swimming path of the control group at 12th time point of step 1 (nominal concentrations of glyphosate suspension was 0 mL·L^−1^); (**b**) swimming path of glyphosate treatment (treatment group) at 12th time point of step 1 (nominal concentrations of glyphosate suspension was 63.03 mL·L^−1^); (**c**) swimming path of glyphosate treatment (treatment group) at 2nd time point of step 2 (nominal concentrations of glyphosate suspension was 63.03 mL·L^−1^). Paths measured with Digimizer (MedCalc Software, Version 4.6.1) on a Windows platform. The yellow lines are fish swimming path.

**Figure 5 vetsci-08-00218-f005:**
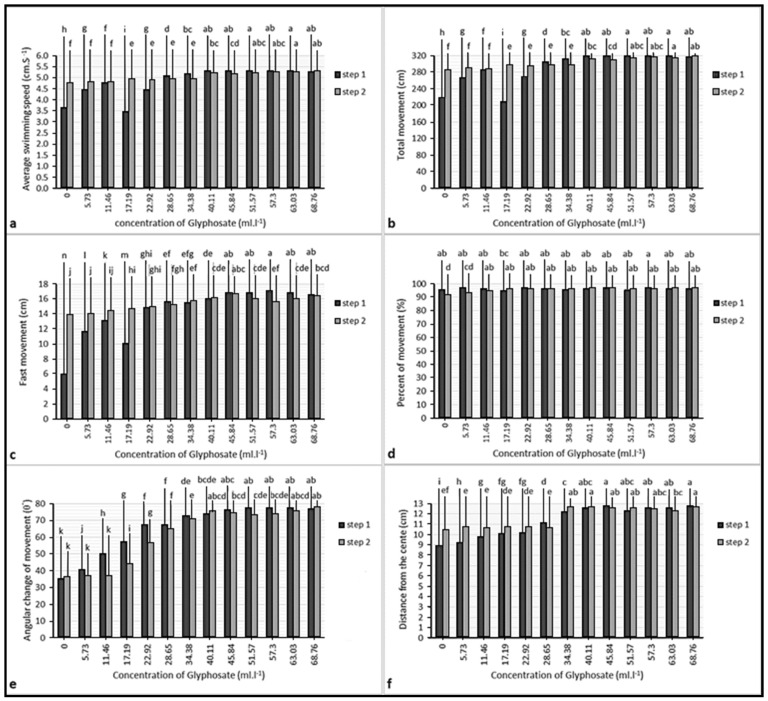
Parameters of swimming pattern of the fish in the control group: (**a**) average speed; (**b**) total movement; (**c**) fast movement; (**d**) percent of movement; (**e**) average angular change of movement; (**f**) distance from the center. Different lowercase letters (a, b, c and else) indicate significant differences between values of the same color columns (*p* < 0.05).

**Figure 6 vetsci-08-00218-f006:**
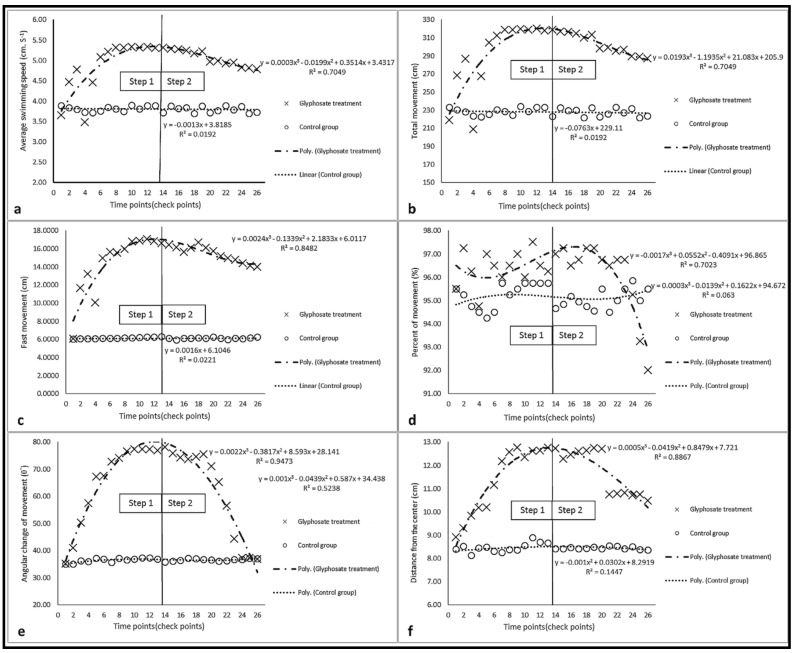
Diagrams of swimming parameters of fish in the test steps: (**a**) average speed; (**b**) total movement; (**c**) fast movement; (**d**) percent of movement; (**e**) average angular change of movement; (**f**) distance from the center.

**Table 1 vetsci-08-00218-t001:** Parameters of swimming patterns of fish, that are used in the present study as characteristics of behavioral responses of common carp (*Cyprinus carpio*) exposed to acute changes of environmental parameters (ammonia and temperature).

Swimming Pattern Parameters	Definition
Average speed swimming (A.S.)	The average speed of the fish in *t* seconds, when the fish move to *X* cm $\left(\bar{V}=\frac{\left(x_{1}+x_{2}+x_{3}+\ldots .+x_{t}\right)}{t}\right)$.
Total movement (T.M.)	Total movement of the fish in *t* seconds: $X_{T}=t\left(\left(x_{1}v_{1}\right)+\left(x_{2}v_{2}\right)+\left(x_{3}v_{3}\right)+\ldots +\left(x_{t}v_{t}\right)\right)$. The movement is the displacement of the fish body by about two-thirds of their body length
Percent movement (P.M.)	Percent of movement is total movement time (*t*) to total time (*T*) multiply by 100 $\left(P.M=\frac{t}{T}\times 100\right)$
Fastest movement (F.M.)	The total distance that fish in that time (1 s) has more than double the average swimming speed.
Average angular change of movement (A.C.)	The angle differences from points t_2_ to t_1_ when the point t_0_ is beginning to move.
Average distance from the center (D.C.)	Average distance of specific region of fish (i.e., fish head) from the center of the test tank in *t* time.

Note: all parameters were selected according to Kane et al. [37] and modified.

**Table 2 vetsci-08-00218-t002:** Gill damages in common carp (*Cyprinus carpio*) after 96 h exposure to commercial formulation of glyphosate (glyphosate Aria 41% SL— suspension).

	Nominal Concentrations (mL·L^−1^)			
Tissue Damages	0	50	100	150
Hyperemia	—	++	+++	+++
Hyperplasia	—	+++	+++	+++
Hypertrophy	—	++	++++	++++
Swollen primary gill	—	++	+++	+++
Secondary lamellar adhesion	—	+++	+++	++++
Hemorrhage	—	++	++	+++
Necrosis	—	+	+++	++++

(-) No gill tissue damage could be seen; (+) there were gill tissue damages from 1 to 3; (++) there were gill tissue damages from 3 to 5; (+++) there were gill tissue damages from 5 to 9; (++++) there were gill tissue damages from 9 to 15.

**Table 3 vetsci-08-00218-t003:** Comparison of the average of swimming parameters of different groups in all the different steps of the test. Parameters were average swimming speed (A.S.), total movement (T.M.), percent movement (P.M.), fastest movement (F.M.), average angular changes of movement (A.C.) and the average distance from the center (D.C.).

	The Test Groups			
Parameters	Control Group		Glyphosate Treatment	
	Step 1	Step 2	Step 1	Step 2
A.S. (cm·s^−1^)	3.82 ± 0.02 ^b^	3.79 ± 0.02 ^b^	4.85 ± 0.18 ^a^	5.06 ± 0.05 ^a^
T.M. (cm)	228.84 ± 1.08 ^b^	227.31 ± 1.26 ^b^	290.87 ± 10.85 ^a^	303.39 ± 3.29 ^a^
P.M. (%)	95.25 ± 0.16 ^b^	95.06 ± 0.12 ^b^	96.38 ± 0.21 ^a^	96.10 ± 0.46 ^a^
F.M. (cm)	6.14 ± 0.02 ^b^	6.11 ± 0.02 ^b^	14.40 ± 0.91 ^a^	15.40 ± 0.25 ^a^
A.C. (θ°)	36.46 ± 0.23 ^b^	36.57 ± 0.12 ^b^	65.46 ± 4.10 ^a^	61.57 ± 4.64 ^a^
D.C. (cm)	8.47 ± 0.06 ^b^	8.44 ± 0.02 ^b^	11.34 ± 0.40 ^a^	11.72 ± 0.27 ^a^

Note: different lowercase letters (^a^ and ^b^) indicate significant differences (*p* < 0.05) between the values in the same row (*p* < 0.05).

## Data Availability

The data presented in this study are available on request from the corresponding author. The data are not publicly available due to proprietary nature of Iranian Ministry of Science Research and Technology.

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
