# Peer review of "Evaluation of Behavioral Changes and Tissue Damages in Common Carp (Cyprinus carpio) after Exposure to the Herbicide Glyphosate"

_vetsci, 2021, doi:10.3390/vetsci8100218_

Round 1
Reviewer 1 Report
The manuscript still needs extensive English corrections which should be done by a native speaker.
In the Material and Methods section, several details are missing:
- Physicochemical parameters: how have these been measured? Are you sure you measured NH3? And for writing the chemical formulas of the parameters, some low letters are needed.
- when adding the glyphosate manually for the range-finding test, was the water mixed also manually or by pumping? The manual mixing may disturb the fish.
- the second experiment was performed with a water flow of what? You should also indicate how many litres per hour entered the tank. If it is too low you cannot expect homogenous buidling up of the test concentrations.
- Figure Caption 1: the glyphosate concentration looks strange
- data analysis: in real-time it would need to work with 30 frames per sec, so I assume the "real-time" should be deleted from this sentence
- video preparation from above means 2D monitoring. So, what was the height of the tanks? This would be interesting to know because of the missing third dimension.
- when you perform an LSD test then you should perform an F test before that. Please report the results from the F test.
Figure 3 and 4 should contain only box plots and the text on each of the axes should be big enough for reading.
The texts on Fig 5 are still unreadable. There is some scaling information on each picture but it is too small for reading.
Fig 6 the graphs are too small and blurred.
In the methods description you mention correlation analyses, but it is not clear what results you obtained from these.
Author Response
Reviewer #1 comments:
- The manuscript still needs extensive English corrections which should be done by a native speaker.
Authors
- The authors thank you for your attention. We are aware that English is not our native language therefore the manuscript was sent on English proofreading.
In the Material and Methods section, several details are missing:
- Physicochemical parameters: how have these been measured? Are you sure you measured NH3? And for writing the chemical formulas of the parameters, some low letters are needed.
Authors
- Done, the pH and temperature, dissolve Oxygen (DO), NH3 concentration and total hardness of water (CaCO3 concentration) were measured by digital soil and substrate pH meter (S500 pro, Aqua Masters, US), dissolve Oxygen meter for aquaculture (HI9147, HANNA Instruments, Slovenia), multiparameter photometers (7100, Palintest, UK) twice a day, respectively. This information was added to the text.
- When adding the glyphosate manually for the range-finding test, was the water mixed also manually or by pumping? The manual mixing may disturb the fish.
- The pesticide was distributed by water circulation inside the aquarium (page 3, line 127).
- The second experiment was performed with a water flow of what? You should also indicate how many liters per hour entered the tank. If it is too low you cannot expect homogenous building up of the test concentrations.
- The experiment was performed based on the LC50 96h concentration of glyphosate. Volume of the test tanks was known and water flow was 416.667 ml.h-1; each step of the behavior test was stopped when exchanged of water were performed (5 liters). The glyphosate concentration in inlet water was 788 ml.L-1 (The LC50 96h concentration of glyphosate).
- Figure Caption 1: the glyphosate concentration looks strange
- The behavior test was a dynamic test; during the test fresh water pumped into tube by water pump and mixed with glyphosate in mixer and then water pumped to test tank by Precision Pump. The concentration of glyphosate in inlet water to the test tank was 68.788 ml.L-1 and water flow was 416 ml.h-1 (Figure 1a). The part b of Figure 1 showed gradient of glyphosate concentration during the test and each step and their time. New information was added to the caption 1.
- data analysis: in real-time it would need to work with 30 frames per sec, so I assume the "real-time" should be deleted from this sentence
- Done, it was deleted
- Video preparation from above means 2D monitoring. So, what was the height of the tanks? This would be interesting to know because of the missing third dimension.
- The height of the cameras from the water surface were 10 cm. information were added to the paper
- When you perform an LSD test then you should perform an F test before that. Please report the results from the F test.
- According to results of test of homogeneity of variance (Levene’s Test), there didn't significant differences between variance of groups (p>0.05). New information was given in the text.
- Figure 3 and 4 should contain only box plots and the text on each of the axes should be big enough for reading.
- Figures were improved
- The texts on Fig 5 are still unreadable. There is some scaling information on each picture but it is too small for reading.
- Figure was improved
- Fig 6 the graphs are too small and blurred.
- Figure was improved
- In the methods description you mention correlation analyses, but it is not clear what results you obtained from these.
- Correlation analyses were mention in order to indicate independence of parameters.
Reviewer 2 Report
By reviewing this manuscript, the author improved this manuscript, I think it reaches to publish on this journal. But, there are still several questions to be resolved well.
1. Your manuscript needs careful editing by someone with expertise in technical English.
2. You should normalize the spelling ‘common carp’ in your manuscript.
3. The high resolution images are also needed.
Author Response
Reviewer #2 comments:
By reviewing this manuscript, the author improved this manuscript; I think it reaches to publish on this journal. But, there are still several questions to be resolved well.
- Your manuscript needs careful editing by someone with expertise in technical English.
- The authors thank you for your attention and your recommendation. The manuscript was edited by proofreader.
- You should normalize the spelling ‘common carp’ in your manuscript.
- Thank you, all of them were replaced
- The high resolution images are also needed.
- All imagines were improved
Reviewer 3 Report
The authors of the manuscript entitled "Evaluation of behavioral changes and tissue damages of common Carp (Cyprinus carpio) after exposure to herbicide-glyphosate" made the corrections suggested by the reviewer. Now I can suggest the publication of the manuscript in the present form by the editorial board of Veterinary Sciences.
Author Response
Reviewer #3 comments:
The authors of the manuscript entitled "Evaluation of behavioral changes and tissue damages of common Carp (Cyprinus carpio) after exposure to herbicide-glyphosate" made the corrections suggested by the reviewer. Now I can suggest the publication of the manuscript in the present form by the editorial board of Veterinary Sciences.
- The authors thank you for your attention and your recommendation
Round 2
Reviewer 1 Report
Dear authors,
the sentence in line 164 looks strange: "Thise information was added to the text." I think it can be omitted.
Moreover, it still not seems that a native speaker has checked the manuscript. For example in line 181/182 it says "The 181 results of the histopathological assay of gills showed in Table 2.", but you should rather write that this is shown in the respective table.
The graphs e.g. in Figure 3 are still not nicely looking. They are not in the correct position (some are overlapping, some are not) and Excel figures are really not the best one for a publication or at least need to be worked on (e.g. less lines in the background, bigger fonds, thick margins or no margins at all?).
For Figure 5 I would still recommend to cut away the whitish text that is unreadable below the tracked lines.
The Figure caption for Fig 5 also shows the units ml L-1 with too many letters in upper cases.
The graphs for Figure 6 should be provided as high resolution files and then transferred to the manuscript at a size that allows reading the equations.
Author Response
Reviewer comments:
- The sentence in line 164 looks strange: "Thise information was added to the text." I think it can be omitted.
Authors
- Done, mentioned sentence was deleted
- Moreover, it still not seems that a native speaker has checked the manuscript. For example in line 181/182 it says "The 181 results of the histopathological assay of gills showed in Table 2.", but you should rather write that this is shown in the respective table.
- We overviewed that mistake, due changes made by Track changes option. It is now corrected (correction is highlighted in green color- lines 194-195). Once again, we apologize.
- The graphs e.g. in Figure 3 are still not nicely looking. They are not in the correct position (some are overlapping, some are not) and Excel figures are really not the best one for a publication or at least need to be worked on (e.g. less lines in the background, bigger funds, thick margins or no margins at all?).
- Done, figure 3 were corrected according to your suggestions.
- For Figure 5 I would still recommend to cut away the whitish text that is unreadable below the tracked lines.
- Done, figure was replaced with improved one.
- The Figure caption for Fig 5 also shows the units ml L-1 with too many letters in upper cases.
- Done, we corrected according to your suggestion.
- The graphs for Figure 6 should be provided as high resolution files and then transferred to the manuscript at a size that allows reading the equations.
- Done, we provided new figure with improved resolution.
This manuscript is a resubmission of an earlier submission. The following is a list of the peer review reports and author responses from that submission.
Round 1
Reviewer 1 Report
The paper describes effects of a commercial glyphosate formulation on common carp. However, the description of the methods are insufficient and the authors claim that the behavioural observations are advantageous compared with the mortality detection. But the manuscript does not show this advantage in numbers.
It is possible that the advantage can be proven, but the description of the material and methods and the results are insufficient at the moment and impair the evaluation of the meaning of the study.
Moreover, the English needs a lot of improvement and this also impairs understanding of all details within the study.
In the following sections, some suggestions for improving the manuscript are listed:
Abstract:
- Does the commercial glyphosate solution contain solvents. The these should have been used in control exposures as well
- gill changes would also be obvious without killing the fish? So not only the behaviour would be a potential non-invasive biomarker.
Introduction
- page 2: illness of whom?
- With respect to the loss of crops: but the amount of crops in general increased? And have the same crops been grown??
- Reaching only 1% of the target organisms seems to be very low. There are surely citations contradicting this.
- The half-life of glyphosate is variable. This has not been taken into account.
- Consumption seems to be the wrong word.
- common carp should not be capitalized and the scientific name only needs to be mentioned once.
- Why to the authors only tend to investigate?
Material & Methods
- The water parameters should be named correctly throughout the manuscript and the units have to be written correctly as well.
- Was the highest pesticide concentration derived from some previous study? And is there some variation for the LC50 between studies?
- Was the tox test performed in a static system?
- how were the glyphosate concentrations added and how was the pesticide distributed in the aquarium?
- The formaldehyde concentration is high and was it buffered? In addition, report thickness of the gill slices and the staining procedures should be described. how was the organ damage analysed?
- For the behavioural assay the study design is unclear. Especially the time stations are not helpful. Use the exact minutes. Maybe a better figure would be helpful here. I doubt that the treatments have these been independent groups? Have all of the analyses been made on one days? For the last experiment – how many fish were left in the group tanks? Were there effects of isolation on the fish behaviour in the small tanks? Report flow rate for the tanks. How can you make sure that the substance is correctly diluted in the tank? Are you sure the glyphosate value was zero after 24 h ? How long was each video and was it analysed as one? The statistics need to be explained more in detail. And 8 fish per group is not really a lot for estimating mortality correlations. The values that have been calculated from the behavioural analyses: explain more clearly and put numbers in brackets for the equations.
- What file format was used for the video analyses. What Adobe software was used exactly? The software was converted? You lose a lot of information if you have only one frame per sec – what is the rationale for your decisions? You claim that different sections have been defined – which sections ? Was it a 2D analyses or 3D? The different correlation analyses should be explained more in detail. Do you assume that a video of 1 min is enough for monitoring opercular movements? And when has this minute been selected? Always at the same time of the experiment? And if the skin colour was estimated – how can you make sure that it is comparable for each tank – the light conditions may be variable and I don`t see any validation of the measurements.
- It is not mentioned if a legal authority gave permission for these experiments.
Results
- The histological pictures are not of good quality
- The footnote of the table 1 is not very clear: does 9 o 15 refer to the number of animals?
- Fig 3: time stations is not clear and the high number of "Aa" is not helpful. And I don`t see a "B", so the "Aa" does not make sense.
- Fig 4: the figures look like Excel graphs. Not very nice and with a high number of different letters. Not helpful for understanding the results. And percent of movement compared to what ?? And the levels on the x axis: how were these concentrations determined? And why the high accuracy of these numbers?
- Fig 5: not good quality and unclear description, e.g. checkpoint? 12th time station? Why not describe the seconds or minutes without this checkpoint things?
- Fig 6: the graphs are too tiny. No one can read this.
- Last section here: how can a speed be compared with a tank area?? And darkening of the skin, increasing movement of operculum, anxiety, jumping out of the water and swimming near the surface have not really been described.
Discussion
There are too many problems with the previous sections so that I cannot judge if the discussion is correct.
Reviewer 2 Report
The study aimed to evaluate behavioral changes and tissue damage in Cyprinus carpio exposed to glyphosate concentrations. Furthermore, to investigate the possible use of the swimming pattern of fish as a behavior
parameter in toxicological studies and environmental quality assessment.
I have some considerations about the manuscript.
Introduction
a) Include the meaning of the abbreviation LC50.
b) Could the authors include studies that use this behavioral analysis model to analyze the effects of herbicides on other fish species?
Materials and Methods
a) Please include the committee approval number for the use of animals.
b) Was the health parameter of the animals obtained ensured? Please include information.
c) Include water quality parameters as a result of the study (Preparing and Toxicity test). I recommend including daily results as mean and standard deviation.
d) Toxicological study: how many samples were collected at each time? Were samples bilateral; Describe about the calculation of LD50.
Results
a) The bar graph in Figure 3 and 4 impairs understanding. It is not possible to verify the exact value of the means and whether there was any statistical difference. Prefer a table.
b) The size of the diagrams shown in Figure 6 makes it difficult to visualize.
Reviewer 3 Report
In this manuscript, the author presented the evaluation of behavioral changes and tissues damages of common carp after exposure to herbicide-glyphosate. I think that it is a good idea. By reviewing this manuscript, however, I think it fails to publish on this journal. There are several reasons as follow:
- Your English is very poor, I cannot understand what you mean in many places of this paper.
- I cannot find your originality and your significance.
- Your design is fuzziness, and I cannot make nothing of it.
- Vajargah et al has done a similar experiment, you do it again, what's your novelty? And the concentrations of LC50 96h measured by you and Vajargah are very different, why do you still say the same in the section of Discussion?
- What are the lethal and sublethal concentrations that you chose for your behavioral experiments?
- Please make sure the spelling ‘common carp’ or ‘common Carp’.
- The p value should be italic.
- In Figure 2, please exchange high resolution images, and the layout of these images is not reasonable.